# Formulations Based on Drug Loaded Aptamer-Conjugated Liposomes as a Viable Strategy for the Topical Treatment of Basal Cell Carcinoma—In Vitro Tests

**DOI:** 10.3390/pharmaceutics13060866

**Published:** 2021-06-11

**Authors:** Anca N. Cadinoiu, Delia M. Rata, Leonard I. Atanase, Cosmin T. Mihai, Simona E. Bacaita, Marcel Popa

**Affiliations:** 1Faculty of Medical Dentistry, “Apollonia” University of Iasi, 700511 Iasi, Romania; jancaniculina@yahoo.com (A.N.C.); leonard.atanase@yahoo.com (L.I.A.); 2Department of Experimental and Applied Biology, NIRDBS—Institute of Biological Research Iasi, 700107 Iasi, Romania; mihai.cosmin.teo@gmail.com; 3Advanced Center for Research and Development in Experimental Medicine (CEMEX), “Grigore T. Popa” Medicine and Pharmacy University of Iasi, 700454 Iasi, Romania; 4Faculty of Machine Manufacturing and Industrial Management, Gheorghe Asachi Technical University of Iasi, D. Mangeron Bld. No. 73, 700050 Iasi, Romania; bsimona77@yahoo.com; 5Academy of Romanian Scientists, 050094 Bucharest, Romania

**Keywords:** liposomal formulation, transdermal delivery, aptamer AS1411, basal cell carcinoma

## Abstract

Topical liposomal drug formulations containing AS1411-aptamer conjugated liposomes were designed to deliver in a sustained way the 5-fluorouracil to the tumor site but also to increase the compliance of patients with basal cell carcinoma. The 5-fluorouracil penetrability efficiency through the Strat-M membrane and the skin irritation potential of the obtained topical liposomal formulations were evaluated in vitro and the Korsmeyer Peppas equation was considered as the most appropriate to model the drug release. Additionally, the efficiency of cytostatic activity for targeted antitumor therapy and the hemolytic capacity were performed in vitro. The obtained results showed that the optimal liposomal formulation is a crosslinked gel based on sodium alginate and hyaluronic acid containing AS1411-aptamer conjugated liposomes loaded with 5-fluorouracil, which appeared to have favorable biosafety effects and may be used as a new therapeutic approach for the topical treatment of basal cell carcinoma.

## 1. Introduction

The 5th most common cancer in humans is non-melanoma skin cancer, in 2018 having over 1 million diagnoses worldwide [1]. Non-melanoma skin cancers are not fatal but can destroy facial sensory organs such as the nose, ear and lips [2,3]. There are some well-established treatments for these non-melanoma skin cancers, such as: curettage and electrodessication; Mohs micrographic surgery; excisional surgery; radiation; cryosurgery; photodynamic therapy; laser surgery and oral medicine [4,5,6]. Unfortunately, these conventional treatments lead to severe inflammation, pain and unpleasant scarring [7]. In cases where the cancer has spread to large areas of the body, topical administration of anticancer drugs is recommended to reduce primarily the cost of surgery and unwanted scars and also to increase patient compliance. Topical administration of anticancer drugs is an easy and effective method to reduce side effects and increase drug targeting and therapeutic benefits [8,9]. At the present, two types of drugs, formulated as creams or ointments, are used almost exclusively: imiquimod (Aldara) [10], for immunotherapy, and 5-fluorouracil (5-FU) (Carac [11], Fluoroplex [12] and Efudex [13]), for skin cancer chemotherapy, with a cure rate of around 80%. The direct treatment with antitumoral drugs has a number of disadvantages as the toxicity of these drugs limits their dose, while rapid clearance from circulation requires large doses in order to be effective [6,8]. It is important to note that the patient needs to rubber these formulations into the tumor once or twice a day for several weeks or longer. This treatment may cause severe redness, irritation, itching, burning, ulceration, scabbing, flaking, pain and swelling or crusting. All these disadvantages limit the feasibility and efficacy of the commercial formulations. Moreover, the percutaneous absorption of 5-FU is its major limiting factor as it penetrates only 1 mm into the skin [6]. 

In order to overcome all these inconvenient, it is necessary to use nanocarriers as drug delivery systems. The main advantage of these systems is that the in vivo fate of the drug is no longer mainly determined by the drug characteristics, but only by the carrier system, which favors a targeted and controlled drug release [14,15]. Moreover, nanotechnology is of interest for skin administration as it can be used to modify the drug permeation/penetration ensuring a direct contact with the stratum corneum [16,17,18]. Specialized literature has reported the use of nanotechnology in reducing the toxicity of 5-FU at therapeutic doses and in improving the 5-FU absorption from the skin surface. Thereby, Sabitha et al. [19] effectively loaded the hydrophilic drug 5-FU on the chitin nanogels to prepare FCNGs, which induced low cytotoxicity on A375 cells, and had the retention in deeper layers of skin 4–5 times greater than the control 5-FU. Additionally, Tiwari et al. [20] designed liposomes containing 5-FU and tretinoin (TTN, a product against acne) and the results showed that liposomes gradually release 5-FU and TTN, being topically safe as pointed out by histological evaluation.

In the literature we can find studies demonstrating the effectiveness of topical treatment with 5-FU on non-melanoma skin cancers: basal cell carcinoma (BCC) [21] and squamous cell carcinoma-related lesions, such as actinic keratosis, Bowen’s disease and keratoacanthoma [22,23,24]. The most common neoplasm is BCC, with rapidly increasing incidence worldwide over recent decades, and is responsible for up to 80% of an estimated 2–3 million annual global incidences of carcinoma [25]. Taking into account the studies presented above, the purpose of our work was to obtain an optimal topical formulation containing AS1411-aptamer conjugated liposomes loaded with 5-FU, which could represent an excellent alternative for the treatment of BCC in the early stages. In our previous article [26], we highlighted the advantages of using liposomes as drug delivery systems but also the possibilities to modify their surface to target cancer cells. From this previous study [26], it appeared that the L4Apt-5FU sample, having a lipoid PC/cholesterol/DSPE-PEG-maleimide ratio of 10/6/1.5, 1.5 mmol AS1411 aptamer, 15 mg/mL 5-FU and a mean diameter of 182 nm, represents the optimal combination in terms of lipid composition, lipids/drug ratio and size. Consequently, three different types of topical formulations (one crosslinked gel, one polymeric physical gel and one cream) were prepared and the previously well-characterized drug-loaded liposomes were incorporated in order to demonstrate their possible application for the efficient treatment of BCC.

## 2. Materials and Methods

### 2.1. Materials

Cholesterol (CHOL), Tris(2-carboxyethyl)phosphine hydrochloride (TCEP•HCl) and 5-fluorouracil (5-FU) were purchased from Alfa Aeser, part of Thermo Fisher Scientific, Kandel, Germany. Lipoid E PC S (Egg Yolk Phosphatidylcholine content: ≥ 96%) (PC) was received as a gift sample from Lipoid GmbH, Ludwigshafen, Germany, DSPE-PEG-maleimide (DSPE-PEG-MAL) from Iris Biotech GmbH, Marktredwitz, Germany DNA aptamer (AS1411-SH) was purchased from Integrated DNA Technologies, BVBA, Leuven, Belgium, chloroform and Triton X were obtained from VWR International Ltd., Lutterworth, United Kingdom. Alginic acid sodium salt (high viscosity), hyaluronic acid sodium salt from *Streptococcus equi*, glycerol, calcium chloride, Strat-M^®^ Membrane (transdermal diffusion test model) 25 mm, isopropanol, MTT powder (3-(4,5-dimethylthiazol-2-yl)-2,5-diphenyltetrazolium bromide), PEG 400 and Pluronic F-108 were purchased from Sigma-Aldrich. Almond oil, Monoi de Tahiti oil, Olliva emulsifying agent, plant-based collagen and Cosgard were acquired from Elemental SRL, Oradea, Romania. Epidermal tissues, small size (0.5 cm^2^), maintenance medium and growth medium were purchased from EPISKIN SA, Lyon, France. Basal cell carcinoma cell line TE 354.T was purchased from ATCC^®^ and the supplies: Dulbecco’s modified growth medium (DMEM), streptomycin and penicillin from Biochrom AG, Germany and fetal bovine serum (FBS) from Sigma, Germany.

### 2.2. Preparation Methods

#### 2.2.1. Preparation of Aptamer-Conjugated Liposomes Loaded with 5-Fluorouracil

The liposomes were prepared by film hydration method followed by sequential extrusion and the AS1411 Aptamer was conjugated on the surface of liposomes as previously described [26]. The sample that was used in this study for the preparation of the liposome-loaded transdermal formulations is L4Apt-5FU-15, abbreviated as L4. This sample has the following composition: 10 mmol PC; 6 mmol Chol; 1.5 mmol DSPE-PEG-Mal; 1.5 mmol AS1411 and was loaded with 15 mg/mL 5-FU. The mean diameter of the chosen sample was determined by dynamic light scattering and its value was 182 ± 27 nm. 

#### 2.2.2. Topical Formulations Preparation

Three different types of topical formulations were prepared in order to incorporate the optimized L4 sample. 

Gel formulation G1 was obtained as a full-interpenetrating alginic acid sodium salt (AG)—hyaluronic acid (HA) network and was prepared by ionic gelation method based on the ionic interaction between calcium cations and carboxylated groups of both polymers. AG and HA solution was prepared by dissolving 1.5% (*w*/*v*) AG and 0.17% (*w*/*v*) HA in 24 mL ultrapure water under continuous magnetic stirring. After complete dissolution of the polymers, 400 µL (0.2 mol/L) of calcium chloride was added drop by drop under continuous stirring, until a homogeneous gel is formed. Then, 5% (*w*/*v*) glycerol, which promotes softness, flexibility and prevents the drying out, was added drop-wise into the formed gel under continuous magnetic stirring. 

Gel formulation G2 is a thermo-reversible gel obtained from Pluronic F-108 in water. Briefly, 3.5 g of Pluronic F-108 was dissolved in 50 mL ultrapure water, under gentle magnetic stirring, at room temperature. After complete dissolution of Pluronic F-108, 5.4 g glycerol was gradually added, under continuous stirring.

The cream C1 is an oil-in-water emulsion in which the oil phase was obtained by mixing 10 g of almond oil and 6.7 g of Monoi de Tahiti oil with 4 g of Olliva emulsifying agent in a heat-resistant container. The aqueous phase was 35 mL of ultrapure water. Both phases were placed in a water bath and heated to 70 °C temperature, with continuous stirring. Once they reached the right temperature, they were removed from the heat source and the two phases were combined and mixed for 3 min using an Ultraturax shaker at 6000 rpm. To accelerate cooling, the composition was placed in a cold water crystallizer and stirred for 3 min. In the cooled composition were gradually added 1.35 g plant-based collagen and 0.45 g Cosgard with thorough mixing after each component. The resulting cream was transferred to a dedicated container. 

Finally, the L4 liposomes suspension (in ultrapure water) at a weight ratio of 1/2 (lipids/formulation) was added to all three previously prepared transdermal formulations.

### 2.3. Characterization 

#### 2.3.1. Rheological Studies of Topical Formulations 

The rheological properties of the prepared formulations (formulations—G1, G2 and C1 and gel formulation containing liposomes—G1–L4, G2–L4 and C1–L4) were analyzed using a modular compact rheometer, Model MCR 302 from Anton Paar, with a cone and plate geometry sensor. The first set of measurements was carried out at 37 °C as a function of the shear rate from 0 to 1000 s^−1^. In a second experiment, the shear rate was kept constant at 50 s^−1^ and the temperature was varied between 10 and 50 °C. Prior to all measurements, the samples were submitted to a shear rate of 50 s^−1^ for 30 s and equilibrated for 2 min at 20 °C in order to standardize their history. 

#### 2.3.2. Transdermal Diffusion Assays across Strat-M Artificial Membrane 

Permeation kinetics assays were performed using a vertical Franz diffusion cell where the two compartments are separated by a transdermal diffusion test model synthetic membrane (Strat-M^®^ membranes, 25 mm diameter). A vertical diffusion cell (VDC) test system model HDT 1000 from Copley was used to maintain a constant temperature, of 32 °C, which represents the normal skin surface temperature, and a constant stirring (600 rpm). The in vitro studies were carried out taking into account the European Medicines Agency guidelines [27,28,29]. The membrane separates the donor compartment, containing either free 5-FU or drug loaded liposomes—L4 or gel formulation with drug loaded liposomes G1–L4, G2–L4 and C1–L4, from the receptor compartment filled with 7 mL collection medium (PBS, pH 7.4). The amount of drug in the donor compartment was 0.4 mg for each tested sample. The diffusion area between the donor and receptor compartments was 1.766 cm^2^. During the entire experiment, the receptor fluid was thoroughly stirred. After elapsed times, 0.2 mL was withdrawn from the receptor compartment solution and an equal volume of fresh prethermostated PBS medium was replaced. Sink conditions were maintained during the experiment and all experiments were performed in triplicate. The drug concentrations in the receiver medium were determined using a Nanodrop One UV–Vis spectrophotometer [30]. The cumulative amount that permeated through the Strat-M membrane per unit area was calculated from the concentration of drug in the receiving medium and plotted as a function of time. 

#### 2.3.3. Mathematical Modelling

When a substance (gel, cream or polymeric matrix) is in contact with water, it retains water and it starts to hydrate from the outside to the inside. Two diffusion fronts appear: one at the interface between the dry and hydrated substance and the second at the interface with water. With the approximations that the diffusivity is constant in time and no erosion takes place, the Fick’s law is given by the Korsmeyer and Peppas equation:(1)$$\frac{M_{t}}{M_{\infty }}=k_{KP}t^{n}.\;$$
where:-***t*** = the time of drug release, -***M_t_*** = the amount of drug delivered at time t, -***M_∞_*** = the total amount of drug delivered after an infinite time interval, -***k_KP_*** = a kinetic constant, a measure of release rate, -***n*** = a diffusional exponent that provides an indication of the mechanism of drug release. 

The value of n up to 0.5 reveals a Fickian diffusion, 0.5–1.0 an anomalous (non-Fickian) transport (i.e., a mixed diffusion and chain relaxation mechanisms) and 1.0 means a Case II transport (zero order) [31]. A value of n greater than 1 reflects the so-called Super Case II-transport [32]. The n values, below 0.5, are associated to drug diffusion through a partially swollen matrix and through water filled pores [33].

#### 2.3.4. In Vitro Evaluation of Topical Formulaons Biocompatibility with Blood Components 

In vitro evaluation of topical formulations biocompatibility with blood components was performed using a spectrophotometric method [34,35], already presented in detail in our previous study [26]. Basal cell carcinoma can cause lesions that often look like open sores, red patches, pink growths, shiny bumps or scars [36,37], so the formulations will inevitably come in contact with blood. We considered that it is of great significance to perform the hemolysis assay because the liposomal formulations will be administered on the skin to the affected area.

All three topical formulations, containing the drug-loaded liposomes G1–L4, G2–L4 and C1–L4 in normal saline solution (2 mL) were added to 2 mL of RBC suspension to obtain a concentration of 0.2 mg/mL. The protocol details are described in the Appendix A section.

#### 2.3.5. The Skin Irritation Potential of the Topical Formulations 

The skin irritation potential of the topical formulations was evaluated using the reconstructed human epidermis SkinEthic™ RHE tissues, which closely mimics the biochemical and physiological properties of the upper parts of the human skin, in its overall design. The experimental procedure followed to determine if the topical formulations obtained induce skin irritation was the Episkin validated protocol for EpiSkinTM Small Model [38]. The first step after receiving the tissues was to transfer them from agarose to growth medium in 6-well plate (1 mL/well) for overnight at 37 ± 2 °C, 5% ± 1% CO_2_ and ≥ 90% humidity. The tissues were transferred the next day to maintenance medium in 24-well plates (300 μL/well) and then were added 10 μL of each topical formulation, in triplicate, to the surface. The sample L4-Apt, G1, G2 and C1 are without 5-FU. The other 5 samples (5-FU; L4Apt-5FU; G1–L4Apt-5FU; G2–L4Apt-5FU; C1–L4Apt-5FU) had the same concentration of 5-FU. After 15 min incubation at room temperature, the tissues were thoroughly rinsed with PBS and transferred into MTT solution 1 mg/mL (300 μL/well; 24-well plates) where they were incubated for 3 h ± 15 min (37 ± 2 °C, 5% ± 1% CO_2_ and ≥ 90% humidity). The formazan extraction was done by immersing the inserts in 750 μL isopropanol and adding another 750 μL on the top of each tissue (extraction from top and bottom of insert). After 2 h with gentle shaking at RT the inserts were perforated to homogenize the formazan extract. Quantification of formazan extract was achieved spectrophotometrically at 570 ± 30 nm. 

#### 2.3.6. Cell Viability Assessment by MTT Method

The human basal carcinoma cell line TE 354.T (ATCC^®^ CRL-7762™) was grown in Dulbecco’s modified growth medium (DMEM, Biochrom AG, Berlin, Germany), containing 10% (*v*/*v*) fetal bovine serum (FBS, Sigma, Germany), 100 μg/mL streptomycin (Biochrom AG, Berlin, Germany), 100 units/mL penicillin (Biochrom AG, Berlin, Germany) and maintained in a humidified atmosphere (approximately 95% air) containing 5% CO_2_ at 37 °C.

The cells were trypsinized in conformity with standard trypsinization procedure with trypsin/EDTA, subsequent being counted and resuspended in 96-well microplates (8 × 103 cells/well), in the same temperature and humidity conditions. After monolayer formation (24 h), the cells were treated for 24 and 48 h with the samples without drug loaded: L4 without 5FU (25–100 µg/mL), C1, G1 and G2 (50–200 µg/mL) and the samples loaded with 5-FU: C1–L4, G1–L4, G2–L4 (50-200 µg/mL) and the 5-FU (25–200 µg/mL). After treatment, the samples were processed by MTT assay [39,40,41], the absorbance being measured at 570 nm using the Biochrom EZ Read 400 microplate automatic reader. The viability of the cells (%) was calculated using formula:(2)$$\mathrm{Cell viability}\left(\%\right)=\frac{\mathrm{Abs}{\;}_{\mathrm{Test}}}{{\mathrm{Abs}}_{\mathrm{Control}}}*100\;$$
where Abs is absorbance.

#### 2.3.7. Apoptosis Assay 

The apoptosis of human basal carcinoma cells (TE 354.T) was investigated at 24 and 48 h after the treatment with the obtained samples by Annexin V-FITC/propidium iodide assay [26,42]. For apoptosis analysis by flowcytometry, the cells were detached, washed, resuspended in binding buffer and stained with AnnexinV-FITC and propidium iodide. The flowcytometric acquisition was performed using Beckman Coulter Cell Lab QuantaSC equipment and appropriate excitation and emission filters were used. The raw data were analyzed with FCSalyzer software.

#### 2.3.8. Statistical Analysis

The statistical significance of cytotoxic activity was analyzed by the unpaired Student’s *t*-test (GraphPad Prism version 8). The values are expressed as mean ± SE of three parallel measurements [43], and the significance different from control was noted by asterisks (* *p* < 0.05, ** *p* < 0.01 and *** *p* < 0.001).

## 3. Results and Discussion

Topical formulations are useful when the BCC is superficial and does not extend very deep into the skin. Schematic representation of the possible action mechanism of the obtained topical formulations to treat BCC is shown in Figure 1.

Liposomes composed of PC, Chol and DSPE-PEG-MAL were prepared, and then aptamer AS1411 that specifically targets tumor cells were conjugated onto the surface of liposomes. Topical formulations containing drug-loaded aptamer functionalized liposomes were manufactured by procedures that do not compromise the performance of the drug and which are reproducible.

### 3.1. Rheological Studies of Topical Formulations 

Figure 2 shows the variation of the apparent viscosity as a function of shear rate at a constant temperature of 37 °C for the G1 gel in the absence and in the presence of liposomes.

Table 1 shows the apparent viscosity values for all the obtained formulations, in the absence and in the presence of liposomes, as a function of temperature.

The study of the rheological properties of these drug-loaded formulations as a function of shear rate and temperature is important because the rheological properties of drug-loaded formulations may affect the release kinetics of the drug.

From Figure 2 it appears that the incorporation of liposomes has no influence on the viscosity of the G1 gel. Moreover, it can also be observed that the apparent viscosity decreases with the increase of the shear rate, for both analyzed samples, which demonstrates a shear thinning behavior. The change of the obstacle and the friction between the polymer chains can produce this tendency of viscosity. Moreover, this shear thinning behavior can be an advantage for the topical application of the obtained formulations. The data reported in Table 1, for shear rate values at 37 °C between 50 and 400 (s^−1^), indicate that the formulation G1 and C1, in the absence or with incorporated liposomes, have a shear thinning effect with an important decrease of the viscosity as a function of the shear rate. In contrast, sample G2 gel has a Newtonian behavior characterized by a constant viscosity in the studied shear rate range. As a function of the temperature at a shear rate of 50 s^−1^, it can be noted that formulations G1 and C1 have almost similar behaviors, with an almost constant viscosity in the temperature range from 20 to 40 °C. On the contrary, the viscosity of the G2 formulation decreased drastically with temperature. Given that G2 is a physical gel obtained by intra and intermolecular hydrogen bonds we can conclude that this decrease may be due to the destruction of hydrogen bonds in the gel structure [37].

### 3.2. In vitro Transdermal Diffusion Assays

The 5-FU permeation profiles across the artificial membrane Strat-M^®^ from free 5-FU solution and drug-loaded liposomes over a time period of 24 h is presented in Appendix A.

Figure 3 presents the 5-FU permeation profiles across the artificial membrane Strat-M^®^ from drug-loaded liposomes incorporated into the topical formulations over a time period of 24 h.

Synthetic membranes for in vitro permeation studies were originally developed to be used as an alternative to animal or human skin models. Some advantages of using a synthetic membrane are: controlled membrane thickness, faster membrane preparation time, low storage space and relatively low cost [44]. In this study, 5-FU-loaded aptamer functionalized liposomes were incorporated in gel/cream formulations and tested in order to demonstrate the potential applicability in cancer treatment.

As can be observed from the Figure 3, the permeability of 5-FU decreased when the drug-loaded functionalized liposomes are incorporated into G1 and C1 formulations and increase when are incorporated into G2 formulations.

### 3.3. Mathematical Modeling

The Korsmeyer Peppas equation was considered the most appropriate to model the drug release because the experimental release kinetics showed that, in the considered time interval, the drug release does not reach the equilibrium phase, the hydration process being in progress, i.e., the swelling degree has not reached the plateau [31,45]. The efficiency of drug release was calculated considering the diffusion surface between compartments equal with 1.766 cm^2^ and approximating that the total amount of drug delivered after an infinite time interval is equal with the amount of loaded drug, i.e., 0.4 mg.

The Korsmeyer Peppas parameters achieved by fitting are presented in Table 2:

The theoretical and experimental release profiles, with correlation factors higher than 0.985, are shown in Figure 4.

One can see from Table 2 that the sample that exhibited a drug release mechanism different from the others is the 5-FU-loaded L4 suspension. In this case, 5-FU diffused only through liposomes membrane whereas for the other ones (G1–L4, G2–L4 gels and C1–L4 cream) an anomalous (non-Fickian) drug transport takes place through simultaneous diffusion and chain relaxation mechanisms.

In addition, the Korsmeyer Peppas equation offers information about the release rate, through k_KP_. The highest value was obtained for the G2–L4 sample, comparable with L4, suggesting that G2 gel accelerates the drug release. For the other two samples, the release rate was much lower, indicating that the G1 gel and C1 cream slowed down the release rate.

### 3.4. In Vitro Evaluation of Topical Formulations Biocompatibility with Blood Components

Results of hemolytic toxicity assay of drug-loaded liposomes and gel/cream formulations after 5 h of incubation, at concentration of 0.2 mg/mL are reported in Appendix A.

The results showed that all tested formulations were found to be lower than 5% hemolysis for all tested concentrations, at all three tested time intervals. 

The hemolysis test was performed in order to determine if the formulations obtained can be used as safe topical formulations even if the surface on which they are applied has lesions that reach to the dermis, the deeper layer of the skin that is very well vascularized. This test can also provide information on the biocompatibility of these topical formulations.

A sample is considered as hemolytic if the hemolytic percentage is above 5% [46]. The hemolysis caused by the tested samples was less than 5% for all tested concentrations, at all tested times, thus proving that the obtained topical formulations are biocompatible and can be safely spread on the skin.

### 3.5. The Skin Irritation Potential of the Topical Formulations

In vitro irritation tests were carried out in order to evaluate the skin irritation potency of the obtained topical formulations by evaluating the cell viability. The reduction of cell viability in treated tissues was compared to negative control (NgC) and expressed as a percentage value.

In Figure 5 are represented the viability percentages obtained after tissue treatments with 10 μL from each topical formulation.

Irritant formulations have been identified by their capacity to decrease cell viability below established threshold levels (below or equal to 50%, for UN GHS Category 2) [47,48]. As illustrated in Figure 5, the sample L4 without 5-FU, G1 and G1–L4, G2 and G2–L4 proved to be non-irritant whereas the samples L4 loaded with 5-FU, C1, C1–L4 and free 5-FU have an irritant effect. These results demonstrate that the obtained gel formulations G1 and G2, with and without immobilized liposomes, are safe to be use in contact with the skin.

### 3.6. Cell Viability Assessment by the MTT Method

In vitro studies on TE 354.T tumor cell cultures were carried out in order to investigate the biocompatibility of the prepared formulations, using several doses ranging from 25 to 200 µg/mL. Moreover, the efficacy of cytostatic activity of these 5-FU-loaded formulations for targeted antitumor therapy was assessed. The samples tested, for 24 and 48 h, were the carriers: L4, C1, G1, G2 and the topical formulations: C1–L4, G1–L4 and G2–L4. The obtained results are presented in Appendix A and Figure 6. 

Treatment for 24 h with the studied samples was materialized by a negligible cytotoxic effect on the TE 354.T tumor cells, the values of the cell viability being included, at the maximum doses administered for each compound, in the range of 77.80–90.74%, the most cytotoxic being the G1 gel, situation, which is also maintained after 48 h of treatment, as illustrated in Appendix A.

Treatment for 24 h with tested formulations resulted in a progressive decrease, in a dose-dependent manner, of cell viability, the tendency becoming more pronounced for the 48 h treatment. Thus, analyzing Figure 6, it was found, at the maximum used doses, a cytotoxic effect of 65.42% for C1–L4, of 56.53% for G1–L4—values above the 50% minimum threshold recommended by in vitro screening programs [49,50,51] and of 40.4% for G2–L4. 

5-FU strongly interfered, especially after 48 h of treatment, with cell viability, having a noticeable cytotoxic effect, of 73.93%, at a maximum dose of 200 µg/mL. As expected, this value exceeds by far the minimum threshold of 50% recommended by in vitro screening programs.

These results are in accordance with the literature data concerning the optimization of the antitumor efficiency of 5-FU by loading in different types of liposomes. Thus, recent studies have formulated 5-FU-loaded pH-sensitive liposomal nanoparticles (pHLNps-5-FU) and have demonstrated their effectiveness against HCT-116 and HT-29 cell lines [52]. These two cell lines treated with pHLNps-5-FU have manifested reduced viability, two or three times lower than that of 5-FU-treated cells. Another recent study, focused on identification of new carriers for drug delivery systems with a high level of biocompatibility, revealed that some lipid nanocapsules (LNCs) loaded with 5–FU had an increased cytotoxic effect on 9L glioma and HTC-116 human colorectal cancer cell line compared to 5-FU or 5-FU modified with lauric acid (5-FU-C12) [53].

Recent studies are focused on the use of two or more antitumor compounds by encapsulating them in colloidal structures that allow the release of drugs in a specific area. Thus, Cosco et al. [54] by coencapsulation of 5-FU and resveratrol in ultradeformable liposomes enhanced the anticancer activity on skin cancer cells both compared to the free drug and compared to single entrapped agents by blocking the cell proliferation in the G1/S phase leading to intensification of resveratrol activity and modulation of the 5-FU effect. Similar results were obtained by Calienni et al. [55] by incorporating 5-FU into ultradeformable liposomes based of soy phosphatidylcholine and sodium cholate (UDL-5FU), the nanoformulation being more toxic on human melanoma than on a human keratinocyte cell line.

### 3.7. Apoptosis Assay 

The treatment, during 48 h (Figure 7B), with the carriers L4 100, G1 200 and G2 200, has induced only a minor effect on cell viability but also on the other correlated parameters (dead, preapoptotic and apoptotic cells). The cell viability in the presence of sample C1 was reduced with a corresponding increase in the frequency of apoptotic cells (Figure 7B). The loading of the carriers with 5-FU determined the reduction of cell viability due to the increase in frequency of preapoptotic and apoptotic cells, with variable amplitudes between the analyzed samples, the largest being induced by the C1–L4 and G1–L4 formulations at a dose of 200 µg/mL.

The treatment with 5-FU at a dose of 200 µg/mL resulted in a significant reduction in cell viability and a considerable increase in the frequency of dead and apoptotic cells, accompanied by a small amplitude reduction in the case of the preapoptotic cells, thereby confirming the expression of the cytotoxic effect through the apoptosis mechanism.

The analysis of the apoptosis process in the case of different topical formulations allowed the recording of variations in the intensity of the apoptotic process, but the effects are transient, and compared to those induced by 5-FU, of smaller amplitude.

A cytotoxic impact similar to that of free 5-FU was recorded after 48 h of incubation with the samples C1–L4 and G1–L4 (Figure 7B). This result may recommend the application of these two samples as potential drug delivery systems for antitumor therapy.

Cosco et al. [54], using the TUNEL assay and evaluation of caspase 3 activity, showed anticancer synergistic effect of 5-FU and RSV coencapsulated in ultradeformable vesicles by promoting a significant apoptotic effect upon Colo-38 human cancer cells, the process being correlated to a high incidence of DNA fragmentation. Moreover, Calienni et al. [55] proved that 5-FU loaded into ultradeformable liposomes composed of soy phosphatidylcholine and sodium cholate (UDL-5FU) determinates a higher level of apoptosis process than free 5FU in the SK-Mel-28 melanoma cell lines.

## 4. Conclusions

This study aimed to found the optimum topical formulation designed to actively target tumor cells and to deliver in a sustained and controlled manner an antitumoral drug to the tumor site in order to increase the compliance of patients with basal cell carcinoma. All tested topical formulations have good compatibility with the bloodstream but not all have demonstrated non-irritating potential, 5FU-loaded liposomes and the cream-like formulation showed a weak irritating potential. The permeability tests of the drug through the artificial membrane Strat M revealed an increase of the permeability of the drug in the case of liposomes incorporated in the G2 gel compared to the free liposomes. The apoptosis tests confirm the good compatibility of the new synthesized complex systems, the degree of tolerability being in the order L4 > C1 = G2 > G1. It can be seen that the incorporation of AS1411 aptamer-functionalized liposomes with the C1 cream and G1 gel exerts a cytotoxic impact closer to that of free 5-FU, which recommends them to be used as efficient drug delivery systems, thus contributing to the improvement of the therapeutic efficacy of antitumor therapy. Finally, it can be admitted that the optimum topical formulation from all points of view is the G1 gel based on biocompatible AG and HA, which appeared to have favorable biosafety effects and may be used as a new therapeutic approach for the treatment of basal cell carcinoma.

## Figures and Tables

**Figure 1 pharmaceutics-13-00866-f001:**
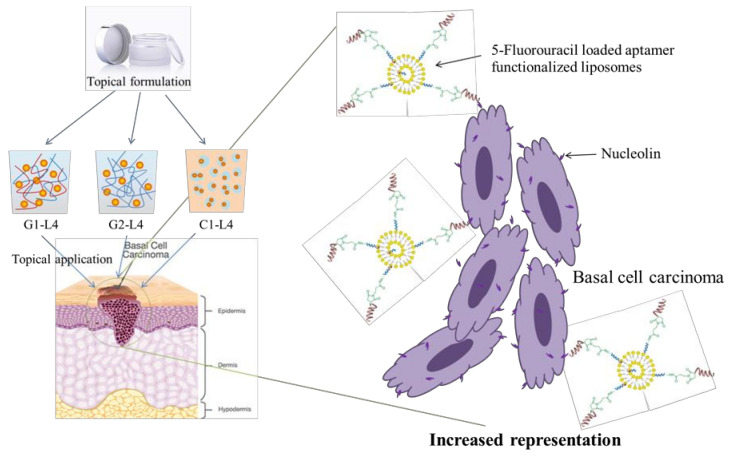
Schematic illustration of the mechanism of action to treat BCC.

**Figure 2 pharmaceutics-13-00866-f002:**
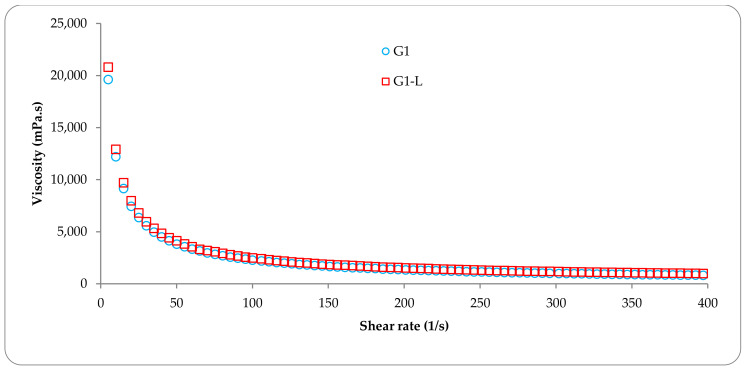
Variation of the apparent viscosity as a function of the shear rate at 37 °C.

**Figure 3 pharmaceutics-13-00866-f003:**
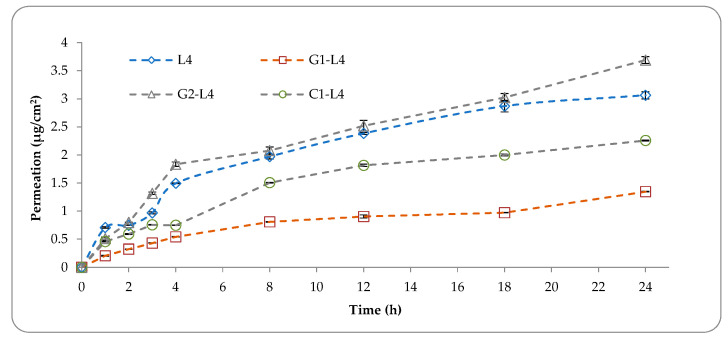
In vitro permeation profiles (μg/cm^2^) of 5-FU across Strat-M membrane in phosphate buffer solution (pH 7.4) from topical formulations with drug loaded liposomes.

**Figure 4 pharmaceutics-13-00866-f004:**
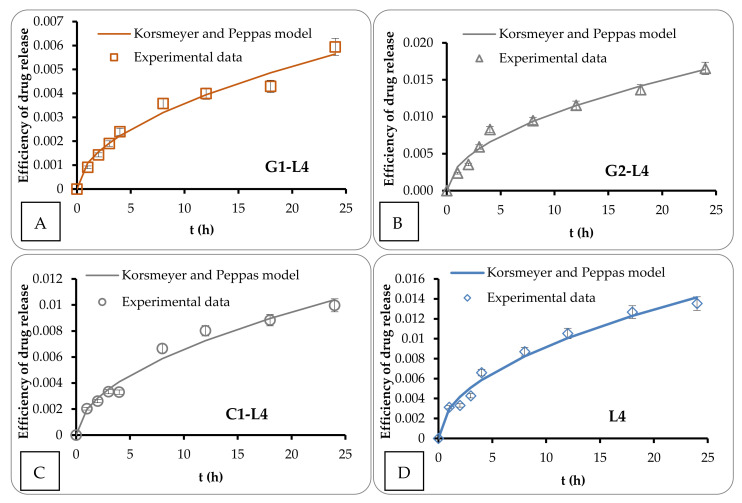
The theoretical and experimental release profiles for G1–L4 (**A**), G2–L4 (**B**), C1–L4 (**C**) and L4 (**D**).

**Figure 5 pharmaceutics-13-00866-f005:**
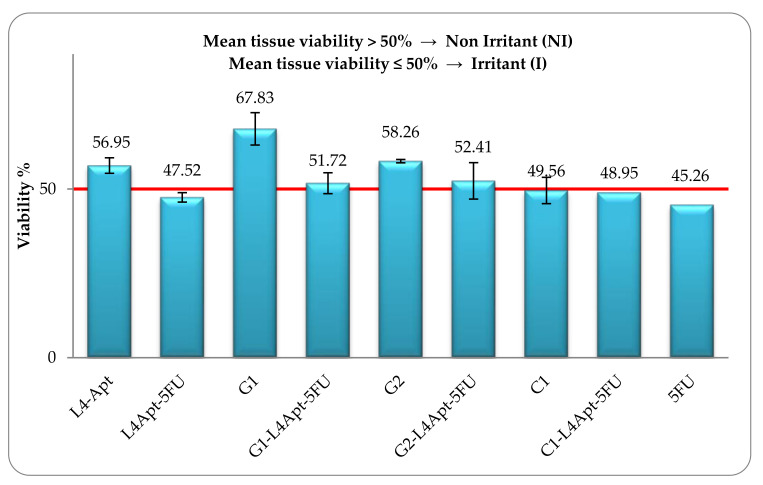
In vitro cell viability in SkinEthic™ RHE tissues treated with different formulations.

**Figure 6 pharmaceutics-13-00866-f006:**
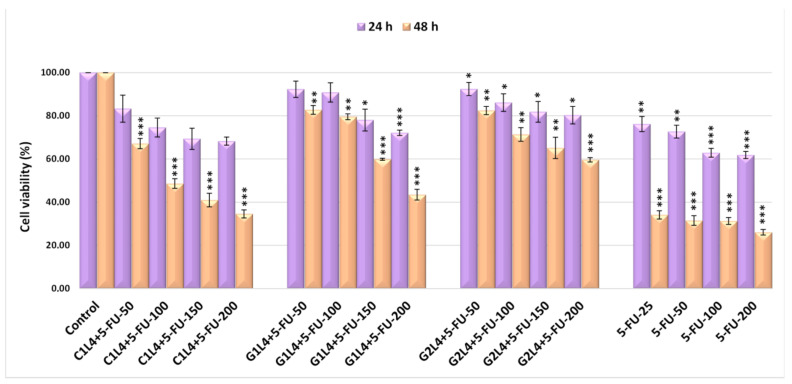
Effect of 24 and 48 h treatment, with different concentrations (µg/mL) of 5-FU and C1-L4, G1-L4 and G2-L4 complexes on the viability of TE 354.T neoplastic cell cultures (significance different from control: * *p* < 0.05, ** *p* < 0.01 and *** *p* < 0.001).

**Figure 7 pharmaceutics-13-00866-f007:**
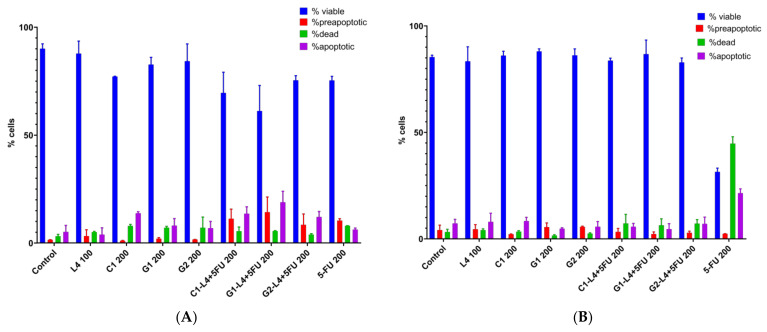
Percentage distribution of the viable, dead, apoptotic and preapoptotic cells after 24 h (**A**) and 48 h (**B**) as quantified by Annexin V-FITC and propidium iodide in apoptosis assay according to every experimental treatment.

**Table 1 pharmaceutics-13-00866-t001:** Apparent viscosity values as a function of shear rate and temperature.

Parameter		Viscosity × 10^−3^ (mPa.s)					
		G1	G1-L4	G2	G2-L4	C1	C1-L4
**Shear rate ^a^** **(1/s)**	**50**	4.12	4.11	0.97	0.91	3.63	3.62
	**100**	2.27	2.25	0.97	0.92	1.82	1.82
	**200**	1.52	1.51	0.98	0.92	0.77	0.78
	**300**	1.18	1.17	0.97	0.91	0.50	0.49
	**400**	0.99	0.98	0.97	0.92	0.38	0.37
**Temperature ^b^** **(°C)**	**20**	6.43	5.91	0.22	0.24	1.55	1.50
	**30**	6.71	6.01	0.12	0.14	1.41	1.28
	**40**	6.33	5.57	0.09	0.09	1.45	1.36

^a^ performed at 37 °C, ^b^ performed at a shear rate of 50 s^−1^.

**Table 2 pharmaceutics-13-00866-t002:** The Korsmeyer Peppas parameters achieved by fitting.

Sample	*k_KP_*	*n*
**G1-L4**	1.088 × 10^−3^	0.518
**G2-L4**	3.245 × 10^−3^	0.510
**C1-L4**	1.986 × 10^−3^	0.521
**L4**	2.963 × 10^−3^	0.492

## Data Availability

The raw/processed data required to reproduce these findings cannot be shared at this time due to technical or time limitations. Data will be made available on request.

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
