# Peer review of "Formulations Based on Drug Loaded Aptamer-Conjugated Liposomes as a Viable Strategy for the Topical Treatment of Basal Cell Carcinoma—In Vitro Tests"

_pharmaceutics, 2021, doi:10.3390/pharmaceutics13060866_

Round 1
Reviewer 1 Report
An interesting original study insìvestigating in vitro the use of a topical liposomial formulation with 5-fluorouracil to henance drug penetration in the skin;
I have some queries:
Please indicate in the statistical analysis section what kind of t-test you did perform (paired, unpaired?); please also specify the statistical program used to calculate statistical significance, its maker and location.
Page 2 line 67 you should add: BCC is the most common neoplasm, with rapidly increasing incidence worldwide over recent decades, and is responsible for up to 80% of an estimated 2–3 million annual global incidences of carcinoma" and cite an article such as: doi: 10.1111/dth.12616.
Thank You
Author Response
Dear Editors and Reviewers:
Thank you for your comments and suggestions regarding our manuscript entitled “Formulations based on drug loaded aptamer-conjugated liposomes as a viable strategy for the topical treatment of basal cell carcinoma - In vitro tests” (ID: pharmaceutics-1237130). These comments are all valuable and very helpful for reviewing and improving our manuscript, as well as the importance of guiding our research. We have responded to the advices and comments accordingly and all the amendments are highlighted in red in the revised manuscript. The main corrections in the paper and the responds to the reviewer’s comments are listed below.
We appreciate Editors/Reviewers’ warm work earnestly and hope that the correction will meet with approval. We are looking forward to your favorable decision.
Sincerely,
Rata Delia Mihaela
Responds to the reviewer’s comments:
- Please indicate in the statistical analysis section what kind of t-test you did perform (paired, unpaired?); please also specify the statistical program used to calculate statistical significance, its maker and location.
R: For data analysis, unpaired Student's t test was used in GraphPad Prism version 8 (San Diego, USA)
- Page 2 line 67 you should add: BCC is the most common neoplasm, with rapidly increasing incidence worldwide over recent decades, and is responsible for up to 80% of an estimated 2–3 million annual global incidences of carcinoma" and cite an article such as: doi: 10.1111/dth.12616.
R: Thank you for your valuable advice. We included the suggested sentence, but to avoid the similarity we have made some adjustments as following: “BCC represents the most common neoplasm, proving rapidly increasing prevalence worldwide over recent decades being responsible for up to 80% of an estimated 2–3 million annual global incidences of carcinoma” [25].
Reviewer 2 Report
The manuscript presents the in vitro characterization of three topical formulations as a potential alternative treatment of basal cell carcinoma. The manuscript fits well into the scope of the Journal. However, the authors have to address some changes to improve their manuscript before publication.
- The introduction should be improved mentioning why the authors propose a novel topical treatment for BCC and not for other types of non-melanoma skin cancers such as SCC. There is evidence that topical treatment with 5-FU has an effect on lesions related to SCC such as actinic keratosis, Bowen’s disease, and keratoacanthoma. Moreover, it could be clarified if the treatment could be suitable or not for metastatic BCC (it is less frequent but exists). In other words, the authors could say that the formulation could be used in the early stages of the disease (if they consider that it is not appropriate for metastatic disease for example).
Additionally, the references are old, they have to be updated to more recent research. The most recent articles they referred to, omitting the author’s previous paper, are from 2014.
- What was the amount of 5-FU incubated for the diffusion studies? Was it the same for all formulations or it was relativized to compare the results? I suggest to include this information in the manuscript to understand the proportion of encapsulated drug which can pass through the membrane after 24 h.
- It is no well clear how the formulations can improve the topical treatment of BCC. The formulations have cytotoxicity against tumoral cells, however, the encapsulation of 5-FU into the conjugated liposomes reduced the drug permeation in comparison to the free drug. Besides, the incorporation of the 5-FU loaded liposomes into the gel or cream reduced still more the drug permeation (except for G2-L4 which increased a bit the permeation in comparison to L4, but it is still considerably less than the free drug).
Regarding this, I have some questions. The artificial membrane Strat-M®, mimics all the skin strata or only the stratum corneum?
Do the authors consider that the conjugated liposomes can penetrate the stratum corneum of the skin? Or do they hypothesize that liposomes release the 5-FU from the skin surface? Have the authors quantified phospholipids or phosphates in the receptor chamber to corroborate the passage of the liposomes? I do these questions because of the size of the liposomes; they are a bit large and I do not know how flexible can they be to overcome the stratum corneum of the skin. If they do not overcome it, it would not make sense to use functionalized liposomes to target cells in the viable epidermis. They would only function as a reservoir of the drug, releasing it over time from the skin surface.
- Why did the authors carry out the hemolysis assays with 0.1 mg/ml of 5-FU (because they diluted in half the formulation 0.2 mg/ml with the sample) if then they use higher concentrations for the cytotoxicity and apoptosis assessments? It would be more relevant to compare the effect on RBC at concentrations with an antitumor effect.
- Regarding the study of skin irritation, are the % of viability reported normalized to the same applied concentration of 5-FU (or the corresponding vehicle without the drug at the same concentration)? The authors have to indicate it in the manuscript because they say that 10 µl of the sample was applied, but I understand that each sample can have a different amount of 5-FU encapsulated. In M&M they say that the liposomal suspension was added at a weight ratio of 1/2 to the gel and cream, therefore, in each formulation, there could be different amounts of the drug. If the % of viability was reported for incubations with different amounts of 5-FU, the figure has to be modified in order to show comparable results. In addition, figure 5 should be improved to allow understanding of which formulations were tested (x-axis), the names are incomplete.
- I suggest the authors improve the discussion for the cytotoxicity assays. They only cited one article from 2015 with 5-FU-liposomes tested on colorectal cancer cell lines, and they mentioned other articles that referred to 5-FU but not for skin cancer and encapsulated in other nanovehicles. In my opinion, it is correct that comparison, but there are articles in which 5-FU was encapsulated into liposomes for topical application against skin cancer too. Therefore, it could be fruitful to also compare with “similar” studies. Some examples: https://doi.org/10.3109/10717544.2014.976891, https://doi.org/10.1016/j.ijpharm.2015.04.056, https://doi.org/10.1007/s13346-017-0469-1
- In table 1, the caption says “Apparent viscosity values at a shear rate of 50 s-1 as a function of temperature”, however, there are also data corresponding to the variation of the shear rate at 37°. The caption has to be corrected.
- Between lines 406 and 410 there is a repeated phrase: “The analysis of the apoptosis process in the case of different topical formulations allowed the recording of variations in the intensity of the apoptotic process”.
- In Figure 7 is very difficult to understand the labels because the letter is too small.
Author Response
Dear Editors and Reviewers:
Thank you for your comments and suggestions regarding our manuscript entitled “Formulations based on drug loaded aptamer-conjugated liposomes as a viable strategy for the topical treatment of basal cell carcinoma - In vitro tests” (ID: pharmaceutics-1237130). These comments are all valuable and very helpful for reviewing and improving our manuscript, as well as the importance of guiding our research. We have responded to the advices and comments accordingly and all the amendments are highlighted in red in the revised manuscript. The main corrections in the paper and the responds to the reviewer’s comments are listed below.
We appreciate Editors/Reviewers’ warm work earnestly and hope that the correction will meet with approval. We are looking forward to your favorable decision.
Sincerely,
Rata Delia Mihaela
Responds to the reviewer’s comments:
- The introduction should be improved mentioning why the authors propose a novel topical treatment for BCC and not for other types of non-melanoma skin cancers such as SCC. There is evidence that topical treatment with 5-FU has an effect on lesions related to SCC such as actinic keratosis, Bowen’s disease, and keratoacanthoma. Moreover, it could be clarified if the treatment could be suitable or not for metastatic BCC (it is less frequent but exists). In other words, the authors could say that the formulation could be used in the early stages of the disease (if they consider that it is not appropriate for metastatic disease for example).
Additionally, the references are old, they have to be updated to more recent research. The most recent articles they referred to, omitting the author’s previous paper, are from 2014.
R: Thank you for your valuable advice.
In the Introduction section, some phrases have been added in which the effectiveness of 5-FU on SCC is mentioned. We have chosen the basal cell carcinoma (BCC) because this type of neoplasm is the most common type of skin cancer in the world that frequently appear after skin exposure to the sun. This cell line is more adequate because we have intended to evaluate the effect of our liposome formulation containing 5-FU on basal cell carcinoma due to its election for skin application. In contrast, the SCC cell line is used on oral/nasal mucosa studies regarding cancer therapy, not proper for epidermal usage. The reviewer's advice was taken into account and more recent studies were added.
It has been specified that this formulation can be used in the early stages. For the research in this paper we did not consider the treatment of metastases, this being a preclinical study in order to establish an optimal doses range to obtain the most appropriate chemotherapeutic response with subsequent applicability in vivo.
- What was the amount of 5-FU incubated for the diffusion studies? Was it the same for all formulations or it was relativized to compare the results? I suggest to include this information in the manuscript to understand the proportion of encapsulated drug which can pass through the membrane after 24 h.
R: The amount of 5-FU was the same for all tested formulations. In the manuscript was added the following sentence: “The amount of drug in the donor compartment was 0.4 mg for each tested sample.”
- It is no well clear how the formulations can improve the topical treatment of BCC. The formulations have cytotoxicity against tumoral cells, however, the encapsulation of 5-FU into the conjugated liposomes reduced the drug permeation in comparison to the free drug. Besides, the incorporation of the 5-FU loaded liposomes into the gel or cream reduced still more the drug permeation (except for G2-L4 which increased a bit the permeation in comparison to L4, but it is still considerably less than the free drug).
R: The purpose of 5-FU encapsulation was to slow the absorption of the delayed-release drug by eluting the toxic side effects of 5-FU. The reactivity of the cells to the tested compounds varied depending on the type of topical formulation, being at 200 µg/mL similar with response obtained at free 5-FU treatment.
Regarding this, I have some questions. The artificial membrane Strat-M®, mimics all the skin strata or only the stratum corneum?
R: According to suppliers, “the multilayered structure of Strat-M™ membrane matches that of human skin. Total membrane thickness: approximately 300 μm. Strat-M™ is constructed of two layers of polyethersulfone (PES, more resistant to diffusion) on top of one layer of polyolefin (more open and diffusive). These polymeric layers create a porous structure with a gradient across the membrane in terms of pore size and diffusivity. The porous structure is impregnated with a proprietary blend of synthetic lipids, imparting additional skin-like properties to the synthetic membrane.” (https://www.merckmillipore.com/RO/ro/product/Strat-M-Membrane-Transdermal-Diffusion-Test-Model-25mm,MM_NF-SKBM02560?ReferrerURL=https%3A%2F%2Fwww.google.com%2F&bd=1#anchor_TI )
Do the authors consider that the conjugated liposomes can penetrate the stratum corneum of the skin? Or do they hypothesize that liposomes release the 5-FU from the skin surface? Have the authors quantified phospholipids or phosphates in the receptor chamber to corroborate the passage of the liposomes? I do these questions because of the size of the liposomes; they are a bit large and I do not know how flexible can they be to overcome the stratum corneum of the skin. If they do not overcome it, it would not make sense to use functionalized liposomes to target cells in the viable epidermis. They would only function as a reservoir of the drug, releasing it over time from the skin surface.
R: Liposomes did not pass through the Strat-M™ membrane. We hypothesize that liposomes have remained in the superficial layers of the membrane where basal cell carcinoma is located. Additional tests to determine how many layers were crossed by liposomes were not performed. The funding received did not allow us to perform ex vivo or in vivo tests but we intend to continue these studies as soon as we receive another funding.
- Why did the authors carry out the hemolysis assays with 0.1 mg/ml of 5-FU (because they diluted in half the formulation 0.2 mg/ml with the sample) if then they use higher concentrations for the cytotoxicity and apoptosis assessments? It would be more relevant to compare the effect on RBC at concentrations with an antitumor effect.
R: The concentration 0.2mg/ml was obtained after the addition of the samples. The phrase in the methodology section was modified: “All three topical formulations, containing the drug-loaded liposomes G1-L4, G2-L4, C1-L4 in normal saline solution (2 mL) were added to 2 mL of RBC suspension to obtain a concentration of 0.2 mg/mL.
- Regarding the study of skin irritation, are the % of viability reported normalized to the same applied concentration of 5-FU (or the corresponding vehicle without the drug at the same concentration)? The authors have to indicate it in the manuscript because they say that 10 µl of the sample was applied, but I understand that each sample can have a different amount of 5-FU encapsulated. In M&M they say that the liposomal suspension was added at a weight ratio of 1/2 to the gel and cream, therefore, in each formulation, there could be different amounts of the drug. If the % of viability was reported for incubations with different amounts of 5-FU, the figure has to be modified in order to show comparable results. In addition, figure 5 should be improved to allow understanding of which formulations were tested (x-axis), the names are incomplete.
R: 4 sample L4-Apt, G1, G2 and C1 are without 5-FU. The other 5 samples had the same concentration of 5-FU. The figure 5 has been modified to see the names of the tested samples (x-axis)
- I suggest the authors improve the discussion for the cytotoxicity assays. They only cited one article from 2015 with 5-FU-liposomes tested on colorectal cancer cell lines, and they mentioned other articles that referred to 5-FU but not for skin cancer and encapsulated in other nanovehicles. In my opinion, it is correct that comparison, but there are articles in which 5-FU was encapsulated into liposomes for topical application against skin cancer too. Therefore, it could be fruitful to also compare with “similar” studies. Some examples: https://doi.org/10.3109/10717544.2014.976891, https://doi.org/10.1016/j.ijpharm.2015.04.056, https://doi.org/10.1007/s13346-017-0469-1
R: Discussions for cytotoxicity tests have improved according to the reviewer's suggestions
- In table 1, the caption says “Apparent viscosity values at a shear rate of 50 s-1 as a function of temperature”, however, there are also data corresponding to the variation of the shear rate at 37°. The caption has to be corrected.
R: As noted by the reviewer, not all parameters were specified in Table 1. The legend and Table 1 have been corrected
- Between lines 406 and 410 there is a repeated phrase: “The analysis of the apoptosis process in the case of different topical formulations allowed the recording of variations in the intensity of the apoptotic process”.
R: We removed the following sentence: “The analysis of the apoptosis process in the case of different topical formulations al-lowed the recording of variations in the intensity of the apoptotic process.”
- In Figure 7 is very difficult to understand the labels because the letter is too small.
R: Figure 7 has been modified

Round 2
Reviewer 1 Report
The authors responded to all queries. The paper is publishable.